# Flame-made ternary Pd-In$_2$O$_3$-ZrO$_2$ catalyst with enhanced oxygen vacancy generation for CO$_2$ hydrogenation to methanol

Thaylan Pinheiro Araújo[1], Cecilia Mondelli [1], Mikhail Agrachev[2], Tangsheng Zou[1], Patrik O. Willi[1], Konstantin M. Engel[1], Robert N. Grass[1], Wendelin J. Stark[1], Olga V. Safonova [3], Gunnar Jeschke [2], Sharon Mitchell[1] & Javier Pérez-Ramírez [1]✉

Palladium promotion and deposition on *monoclinic* zirconia are effective strategies to boost the performance of bulk In$_2$O$_3$ in CO$_2$-to-methanol and could unlock superior reactivity if well integrated into a single catalytic system. However, harnessing synergic effects of the individual components is crucial and very challenging as it requires precise control over their assembly. Herein, we present ternary Pd-In$_2$O$_3$-ZrO$_2$ catalysts prepared by flame spray pyrolysis (FSP) with remarkable methanol productivity and improved metal utilization, surpassing their binary counterparts. Unlike established impregnation and co-precipitation methods, FSP produces materials combining low-nuclearity palladium species associated with In$_2$O$_3$ monolayers highly dispersed on the ZrO$_2$ carrier, whose surface partially transforms from a *tetragonal* into a *monoclinic*-like structure upon reaction. A pioneering protocol developed to quantify oxygen vacancies using in situ electron paramagnetic resonance spectroscopy reveals their enhanced generation because of this unique catalyst architecture, thereby rationalizing its high and sustained methanol productivity.

Indium oxide (In$_2$O$_3$) has emerged as an attractive catalyst with the potential to realize sustainable methanol production via CO$_2$ hydrogenation (CO$_2$ + 3H$_2$ ↔ CH$_3$OH + H$_2$O), owing to its intrinsic high selectivity[1–3]. Detailed mechanistic studies revealed that oxygen vacancies generated under reaction conditions are essential to form a catalytic ensemble that activates CO$_2$ and heterolytically splits H$_2$, and favoring methanol generation over carbon monoxide, the latter forming through the parasitic reverse water-gas shift RWGS reaction (CO$_2$ + H$_2$ ↔ CO + H$_2$O)[3]. Nonetheless, because H$_2$ activation is energetically demanding and limits methanol synthesis over bulk In$_2$O$_3$, there has been great incentive to improve its performance. The most prominent strategies are through deposition on a carrier or introducing a metal promoter[1,4–15]. Among a wide number examined,

zirconium oxide (ZrO$_2$) stood out as an exceptional carrier, particularly the *monoclinic* (*m*) polymorph, boosting methanol space-time yield (*STY*) by 17 and 9-fold per gram of indium compared to unsupported and *tetragonal* (*t*) ZrO$_2$-supported In$_2$O$_3$-based catalysts, respectively, while also securing a stable performance over 1000 h[1,4,9,10]. The extraordinary behavior of In$_2$O$_3$/*m*-ZrO$_2$ was attributed to the enhanced CO$_2$ adsorption capacity granted by *m*-ZrO$_2$ and, more importantly, the creation of additional oxygen vacancies, which was suggested to arise due to distinct phenomena[4,9,10]. For instance, the formation of a solid solution of indium and zirconium oxide at the carrier surface and strain generated upon epitaxial growth of In$_2$O$_3$ on *m*-ZrO$_2$ owing to a delicate mismatch between their lattices, which is less prominent when using the *t* polymorph[4,9]. As expected, impregnation methods proved

[1]Institute of Chemical and Bioengineering, Department of Chemistry and Applied Biosciences, ETH Zurich, Vladimir-Prelog-Weg 1, 8093 Zurich, Switzerland. [2]Laboratory of Physical Chemistry, Department of Chemistry and Applied Biosciences, ETH Zurich, Vladimir-Prelog-Weg 2, 8093 Zurich, Switzerland. [3]Paul Scherrer Institute, Forschungsstrasse 111, 5232 Villigen, Switzerland. ✉e-mail: jpr@chem.ethz.ch

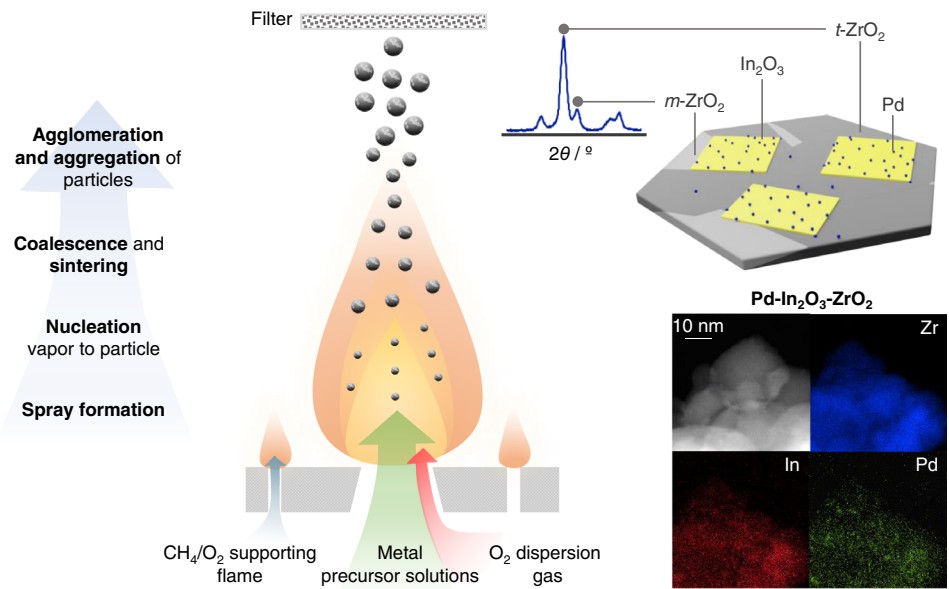

**Fig. 1 | One-step approach to prepare ternary Pd-In$_2$O$_3$-ZrO$_2$ catalysts.** Scheme of the flame spray pyrolysis (FSP) method for the preparation of ternary Pd-In$_2$O$_3$-ZrO$_2$ catalysts (exemplified) and reference binary systems.

crucial in this context, since co-precipitation favors unwanted incorporation of indium into the bulk of ZrO$_2$. This tendency leads to detrimental effects as undesired $t$-ZrO$_2$ is readily formed and stabilized, and less active phase is available at the catalyst surface[4,9,10].

Alternatively, metal promotion has also been extensively explored to boost the activity of In$_2$O$_3$, with palladium offering unparalleled performance enhancement, particularly when anchored to the oxide surface in the form of low-nuclearity species (Pd-In$_2$O$_3$, *STY* of *ca.* 1 g$_{MeOH}$ h$^{-1}$ g$_{cat}$$^{-1}$ over 500 h)[5–7,11–15]. In this case, co-precipitation was key to ensure maximal performance as it enables a stable atomic dispersion of palladium, which is unattainable by impregnation[11]. Unlike large clusters, atomically-dispersed palladium species not only maximize homolytic H$_2$ splitting and oxygen vacancy formation on the In$_2$O$_3$ surface, but do so while minimizing CO formation, thereby unlocking pratically viable methanol productivities[11]. For a prospective industrial process, however, this catalyst formulation can be optimized as a significant amount of palladium and indium, both scarce elements, are inaccessible in the bulk phase. In this regard, we conceived that efficiently integrating palladium promotion and supporting on ZrO$_2$ into a ternary Pd-In$_2$O$_3$-ZrO$_2$ system should grant superior performance with improved metal utilization. Still, this is a challenging task as accurate control over the catalyst architecture (for instance, zirconia polymorph and location and speciation of palladium and indium) is deemed crucial and established routes used for producing binary In$_2$O$_3$/$m$-ZrO$_2$ and Pd-In$_2$O$_3$ systems are likely unsuited to realize it, due to the aforementioned synthetic constraints. Additionally, while critically important to guide catalyst assembly and rationalize its impact on performance, quantification of oxygen vacancies on In$_2$O$_3$-based systems remains elusive. Their assessment predominantly focuses on confirming their formation, which has been demonstrated using electron paramagnetic resonance, nuclear magnetic resonance, Raman, and X-ray photoelectron spectroscopies[1,4,6,9–11]. However, to our knowledge, no attempt has been made to quantify the effects of the catalyst composition and architecture on the oxygen vacancy density. Therefore, the rational design of tailored ternary Pd-In$_2$O$_3$-ZrO$_2$ systems calls not only for alternative synthesis platforms, but also the development of experimental protocols to quantify oxygen vacancies.

Herein, we apply flame spray pyrolysis (FSP) to produce a highly efficient ternary Pd-In$_2$O$_3$-ZrO$_2$, which reaches an outstanding *STY* of

*ca.* 1.3 g$_{MeOH}$ h$^{-1}$ g$_{cat}$$^{-1}$ in CO$_2$-based methanol synthesis. FSP stands out as an appealing method, since it offers a straightforward way to manufacture in a single-step approach multicomponent nanostructures with improved surface area, resistance to sintering, and controlled dispersion and location of carried metallic and oxide phases[16–19]. Indeed, contrary to impregnation and co-precipitation methods, FSP fosters the formation and synergic interplay between low-nuclearity palladium species and In$_2$O$_3$ platelets, which are supported on the surface of a dynamic mixture of $t$- and $m$-ZrO$_2$ particles, as unveiled by in-depth characterization. Kinetic analyses show how this unique catalyst architecture favors methanol production by curtailing the RWGS reaction. Lastly, through extensive investigation using ex situ and in situ electron paramagnetic resonance spectroscopy, we address the long-standing challenge of characterizing and quantifying oxygen vacancies on In$_2$O$_3$-based catalysts, thereby advancing understanding of this critical performance descriptor. Our study opens a new chapter for In$_2$O$_3$-catalyzed CO$_2$ hydrogenation to methanol, reshapes our understanding of the effect of ZrO$_2$ polymorphs, and showcases FSP as a powerful platform for engineering complex heterogeneous catalytic systems for diverse applications.

## Results and discussion
### Impact of synthesis method and promoter content on performance

Flame spray pyrolysis (FSP, Fig. 1) was selected as a one-pot approach to produce a ternary Pd-In$_2$O$_3$-ZrO$_2$ catalyst targeting atomic distributions of palladium and indium on zirconia. Additionally, this method was applied to prepare In$_2$O$_3$, In$_2$O$_3$-ZrO$_2$, Pd-ZrO$_2$, and Pd-In$_2$O$_3$ catalysts for comparative purposes while other reference materials were attained by optimized wet impregnation (WI) and co-precipitation (CP) protocols available elsewhere[4,6,11]. Details on the synthesis and notation of the catalysts are provided in the Supplementary Methods and Supplementary Tables 1 and 2. The nominal palladium and In$_2$O$_3$ contents were accurately reached for all systems, as confirmed by XRF and ICP-OES measurements (Supplementary Table 2). The catalysts were evaluated in CO$_2$ hydrogenation to methanol at 553 K, 5 MPa, H$_2$/CO$_2$ = 4 and 48000 cm$^3$ h$^{-1}$ g$_{cat}$$^{-1}$ for 50 h. The ternary 0.75Pd-5In$_2$O$_3$-ZrO$_2$ (10) catalyst prepared by FSP outperformed all other In$_2$O$_3$-based systems and most previously

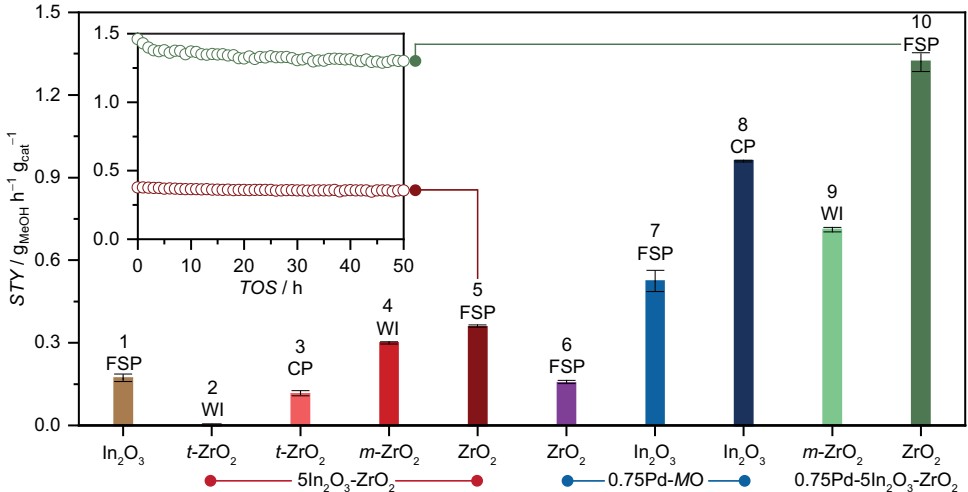

**Fig. 2 | Comparative performance of In₂O₃-based catalysts.** Methanol space-time yield (*STY*) during CO₂ hydrogenation over ternary Pd-In₂O₃-ZrO₂ catalyst prepared by FSP with benchmark In₂O₃-based catalysts and Pd-ZrO₂ serving as reference (WI-wet impregnation, and CP-coprecipitation). Averaged values measured over 50 h on stream are presented with their corresponding error bars. The number preceding the acronym in the catalysts' codes indicates the nominal loading in wt.%, unless otherwise stated. Inset shows the *STY* of selected systems over time-on-stream, *TOS*. Reaction conditions: $T = 553$ K, $P = 5$ MPa, $H_2/CO_2 = 4$, and $GHSV = 48,000$ cm³ h⁻¹ g_cat⁻¹.

reported materials studied under comparable conditions (Fig. 2 and Supplementary Fig. 1 and Table 3), attaining a stable methanol space-time yield (*STY*) of 1.3 g_MeOH h⁻¹ g_cat⁻¹ for 50 h on stream, which is 40% higher than that displayed by its ternary counterpart synthesized by WI (9), because of its higher CO₂ conversion ($X_{CO2}$) and methanol selectivity ($S_{MeOH}$). Additionally, its *STY* is significantly superior to reference binary Pd-In₂O₃ systems (by 25 and 65% for CP (8)[11] and FSP (7), respectively), even though these binary materials contain much more indium than the ternary counterpart (99.25 *versus* 5 wt.%), hinting that ZrO₂ greatly contributes to improve methanol productivity, particularly because it does so while requiring a significantly lower content of indium, which like palladium is also a scarce and expensive material. In contrast, Pd-ZrO₂ (6) obtained through FSP mainly produced CO, as expected from the reported RWGS activity of palladium nanoparticles (Supplementary Figs. 1 and 2)[11,20]. Interestingly, similar to Pd-In₂O₃-ZrO₂, FSP was also instrumental to produce in a single step a In₂O₃-ZrO₂ catalyst (5) with improved performance. In fact, while its methanol productivity is slightly better in comparison to the benchmark system (4) comprising In₂O₃ wet-impregnated on *m*-ZrO₂ (*ca.* 0.45 versus 0.36 g_MeOH h⁻¹ g_cat⁻¹), this catalyst totally surpassed those prepared by CP (3) and WI when using *t*-ZrO₂ as the carrier (2), with the latter being virtually inactive (Fig. 2 and Supplementary Figs. 1 and 3). The inferior performance of these catalysts is in line with previous reports[4,9,10] and further stresses the critical impact of the synthesis method on the performance of binary and ternary systems based on In₂O₃ and ZrO₂. In addition to palladium, a ternary catalyst using platinum as promoter was also produced by FSP (1Pt-5In₂O₃-ZrO₂) which attained inferior methanol *STY* compared to 0.75Pd-5In₂O₃-ZrO₂ (1.0 versus 1.3 g_MeOH h⁻¹ g_cat⁻¹) but reasonable stability, showcasing that FSP is a versatile platform to manufacture other metal-promoted In₂O₃-ZrO₂ catalysts (Supplementary Fig. 4). Upon ascertaining the superiority of palladium over platinum, we unveiled the impact of its content by evaluating materials containing 0.5–2 wt% Pd (Supplementary Fig. 5). The initial loading of 0.75 wt% was found to be optimal since methanol productivity was inferior at lower contents, and remained unchanged or slightly diminished at higher amounts. Finally, $S_{MeOH}$ over catalysts prepared by FSP at a constant $X_{CO2}$ (*ca.* 3%, Supplementary Fig. 2) revealed that the Pd-In₂O₃-ZrO₂ system is generally more selective than any of its other binary formulations and especially Pd-ZrO₂ (87 versus 18%), suggesting that palladium most likely interacts more synergistically with In₂O₃, which exerts a strong influence on

its properties by suppressing CO formation when both phases are introduced to ZrO₂ by FSP.

## Impact of the crystalline structure of the zirconia support

To rationalize the remarkable behavior of the Pd-In₂O₃-ZrO₂ system prepared by FSP, this catalyst and reference materials were investigated by in-depth characterization. X-ray diffraction (XRD) patterns of fresh and used Pd-In₂O₃-ZrO₂ (Fig. 3a) showed no reflections corresponding to palladium phases, which are also absent for reference Pd-In₂O₃, Pd-ZrO₂, and other ternary systems containing distinct amount of the promoter (Supplementary Figs. 6 and 7a), indicating the lack of Pd nanoparticles larger than 4 nm. Similar evidence is also gathered for platinum-containing catalysts (Supplementary Figs. 6 and 7a). Additionally, no signals of In₂O₃ were found for ZrO₂-containing catalysts in fresh or used forms, which is in line with previous reports[4,9,21,22] (Fig. 3a and Supplementary Figs. 6 and 7a) and hints at a high dispersion of the active phase on the carrier. Given the exceptional performance, it is surprising that the diffractograms of Pd-In₂O₃-ZrO₂ (FSP) show characteristic reflections of both *t* ((101) plane at 30.0° 2θ) and *m* ((−111) and (111) planes at 28.1° and 31.5° 2θ, respectively) zirconia, a typical feature also present in Pd-ZrO₂, In₂O₃-ZrO₂, and other ternary systems attained by FSP as well as In₂O₃/*t*-ZrO₂ synthesized by WI (Fig. 3a and Supplementary Figs. 6 and 7a). Quantification of the relative amounts of the zirconia polymorphs (Fig. 3b and Supplementary Fig. 7b) revealed that FSP-derived catalysts in fresh and used forms generally contain 65–80 wt.% *t*-ZrO₂. In contrast, In₂O₃/*t*-ZrO₂,WI initially comprises pure *t*-ZrO₂, which, regardless of the precipitating agent (NH₄OH, NaOH, and ethylene diamine) used in its preparation, partially transformed into *m*-ZrO₂ upon wet-impregnation of indium nitrate (38 wt.%), which becomes the predominant phase (70 wt.%) upon reaction (Fig. 3b and Supplementary Fig. 8). In addition, while most zirconia-based systems showed comparable initial specific surface area ($S_{BET}$), with no sign of sintering after use, the $S_{BET}$ of *t*-ZrO₂ carriers considerably decreased upon deposition of In₂O₃ by WI (from 165 to 50 m² g⁻¹, Fig. 3b and Supplementary Figs. 7b, 8, and 9). These findings further corroborate the XRD results and are commonly observed during the *t-m* phase transition since the unit cell of *m*-ZrO₂ is denser than that of *t*-ZrO₂[23,24]. Although only few reports exist for ZrO₂-supported In₂O₃[10], this phenomenon is thermodynamically favored and well-

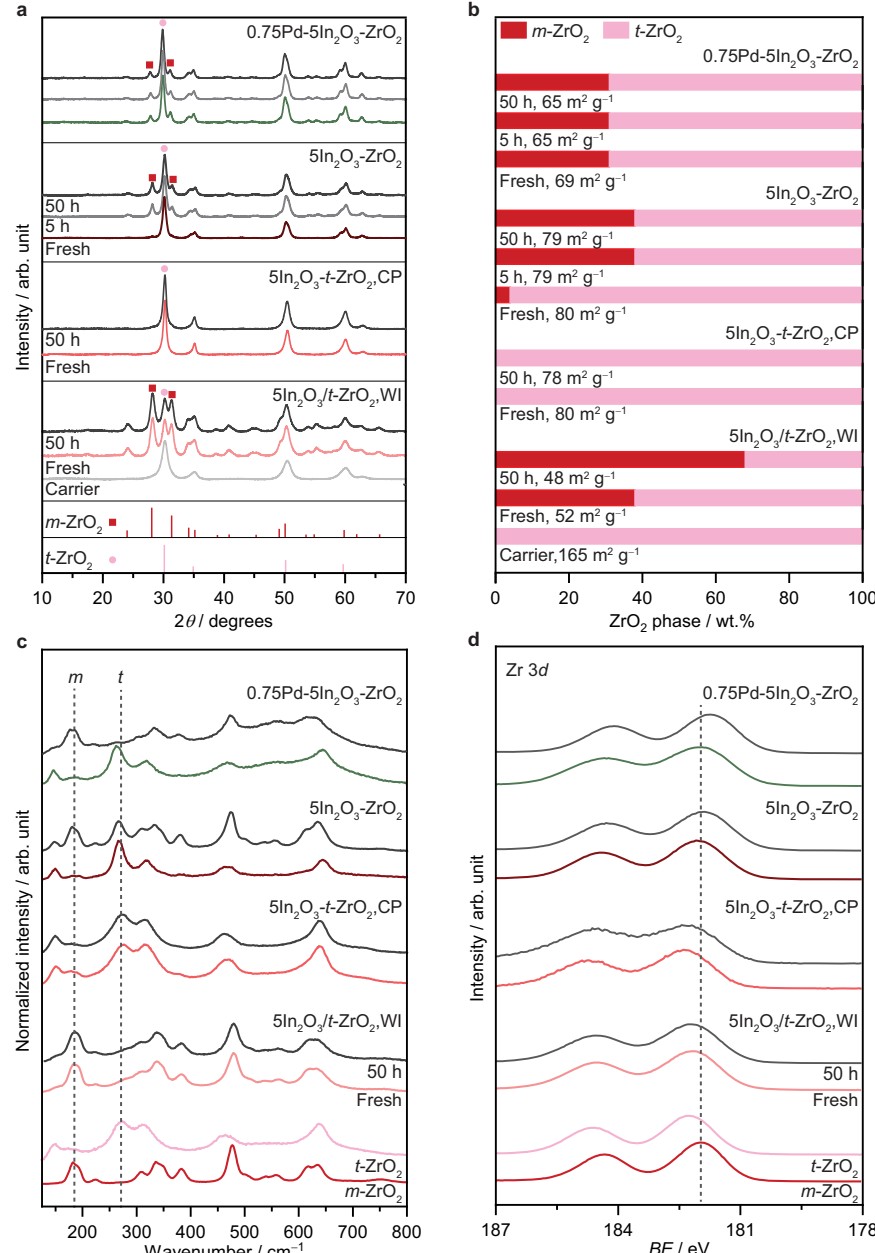

**Fig. 3 | Bulk and surface structure of the zirconia carrier. a** XRD patterns, **b** Zirconia phase composition and BET surface areas ($S_{BET}$), **c** Raman spectra, and (**d**) Zr 3*d* XPS core-level spectra of $In_2O_3$-based catalysts in fresh form and after $CO_2$ hydrogenation for 50 h, with *t*- and *m*-$ZrO_2$ carriers serving as reference in **c** and **d**.

Phase composition shown in **b** was determined from the XRD patterns presented in **a** by applying the reference intensity ratio method, whereas $S_{BET}$ was determined using the $N_2$ isotherm.

documented for undoped and yttrium-doped *t*-$ZrO_2$, being typically triggered by distinct processes[23–27]. The *t-m* transition observed for $In_2O_3/t$-$ZrO_2$,WI systems is likely induced by moisture and thermal treatments using water or its vapor, whereas a combination of synthesis conditions characteristic of FSP such as high temperatures (2300–2500 K) accompanied by very fast cooling rates (*ca.* $10^6$ K $s^{-1}$) facilitates the stabilization of both *m* and *t* phases, thus explaining the presence of both polymorphs in systems attained by this method[17,23–30]. By examining the fresh and used catalysts using UV Raman spectroscopy (325 nm excitation laser, Fig. 3c), additional insights into the zirconia structure at their outmost layer (*ca.* 10 nm) were gathered[10,31]. Interestingly, both Pd-$In_2O_3$-$ZrO_2$ and its binary $In_2O_3$-$ZrO_2$ counterpart are mainly composed of *t*-$ZrO_2$, which restructures upon reaction leading to the formation of a *m*-

$ZrO_2$-enriched surface. The reconstruction is particularly noticeable for Pd-$In_2O_3$-$ZrO_2$, as indicated by the relative change of intensity of vibrational modes characteristic of *t* ($E_g$, 269 $cm^{-1}$) and *m* ($A_g$, 187 $cm^{-1}$) phases[10]. In contrast, the $ZrO_2$ at the surface of $In_2O_3$-*t*-$ZrO_2$,CP and $In_2O_3/t$-$ZrO_2$,WI remained unaltered as tetragonal and monoclinic, respectively. Additional characterization of selected systems using X-ray photoelectron spectroscopy (XPS) revealed a clear shift to lower binding energy (*BE*) in the Zr 3*d* region for used Pd-$In_2O_3$-$ZrO_2$ and $In_2O_3$-$ZrO_2$ catalysts with respect to the pure *m*-$ZrO_2$ phase, which was absent for $In_2O_3$-*t*-$ZrO_2$,CP and $In_2O_3/$ *t*-$ZrO_2$,WI systems (Fig. 3d). This shift in *BE* likely arises from restructuring of the $ZrO_2$ surface and points to a strong interaction between $In_2O_3$ and $ZrO_2$, which possibly reflects an augmented density of oxygen vacancies, and could explain the improved

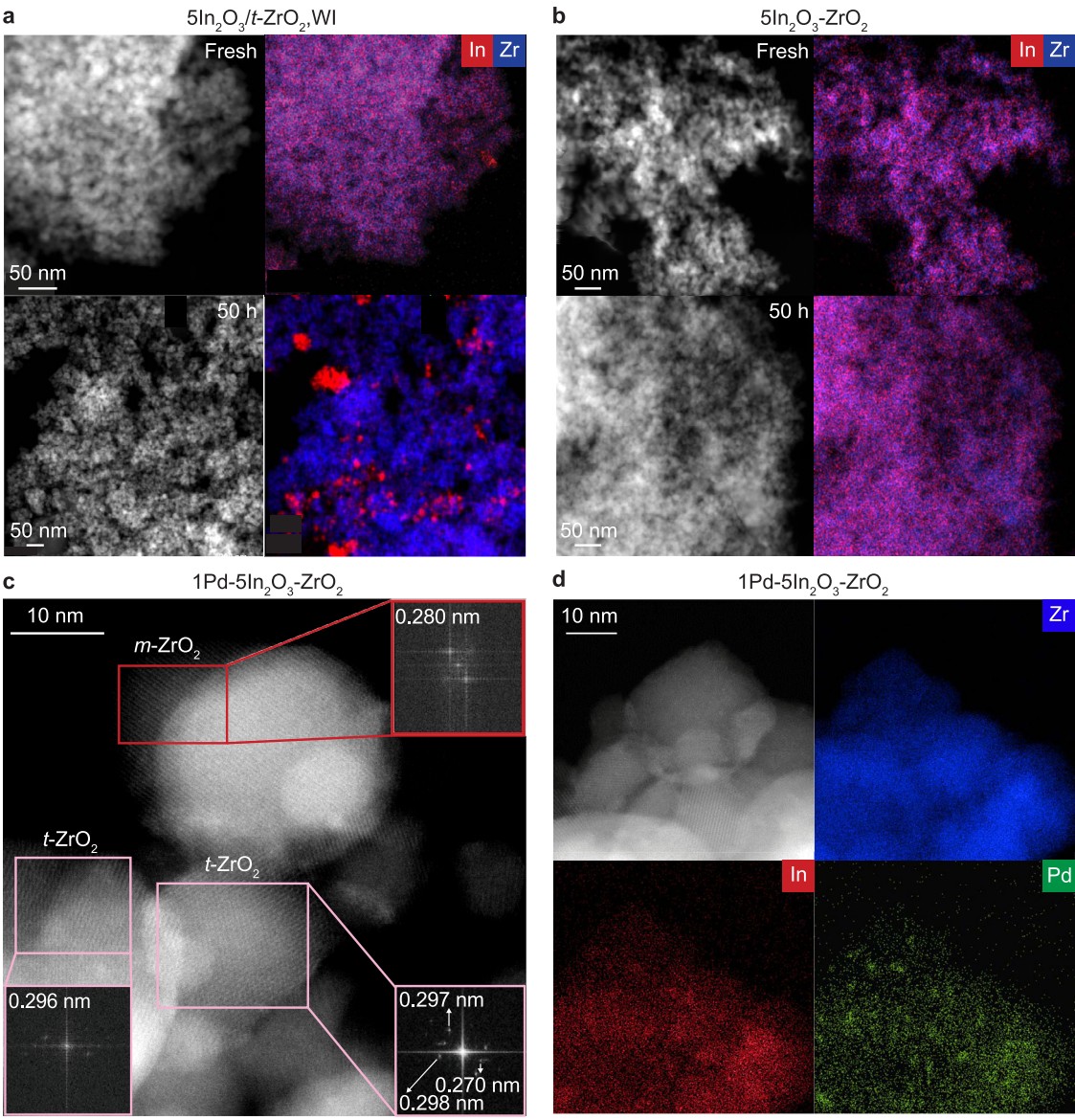

**Fig. 4 | Structural organization of the ternary Pd-In$_2$O$_3$-ZrO$_2$ catalyst. a, b** HAADF-STEM micrographs with EDX maps of zirconia-supported In$_2$O$_3$ catalysts in fresh form and after CO$_2$ hydrogenation for 50 h. **c** AC-STEM images and corresponding EDX maps (**d**) of Pd-In$_2$O$_3$-ZrO$_2$ system prepared by FSP after CO$_2$ hydrogenation for 50 h. Insets in (**c**) show fast Fourier transforms of indicated areas, evidencing the presence of both *m*- and *t*-ZrO$_2$ phases from characteristic lattice distances. Reaction conditions: $T = 553$ K, $P = 5$ MPa, H$_2$/CO$_2 = 4$, and $GHSV = 48,000$ cm$^3$ h$^{-1}$ g$_{cat}^{-1}$.

performance of systems prepared by FSP[32,33]. Overall, contrary to previous studies[9,11,22,32,34], XRD, Raman, and XPS findings indicate that a pure *m*-ZrO$_2$ carrier is not a must to attain a superior methanol productivity, especially for more complex In$_2$O$_3$-ZrO$_2$-based systems.

### Geometric and electronic effects on Pd-In$_2$O$_3$-ZrO$_2$

Analysis by microcopy techniques provided further insights into the architecture of the Pd-In$_2$O$_3$-ZrO$_2$ catalyst and reference systems. High-angle annular dark-field scanning transmission electron microscopy (HAADF-STEM) coupled to energy-dispersive X-ray spectroscopy (EDX) confirmed that In$_2$O$_3$ is well-dispersed on zirconia when deposited by FSP, while it strongly aggregated over all binary catalysts synthesized by WI of *t*-ZrO$_2$ carriers (Fig. 4a, b and Supplementary Fig. 10a, b), which is most likely triggered by the transformation of the latter into *m*-ZrO$_2$, thus explaining the pronounced performance difference among these systems. As expected, no sign of In$_2$O$_3$ agglomeration was evidenced over reference

materials attained through WI of a *m*-ZrO$_2$ carrier and CP (Supplementary Fig. 10a, b). However, some indium is likely incorporated into the bulk of the support in the catalyst obtained by CP, which explains its inferior methanol productivity compared to FSP-prepared systems[4]. The superior features of the latter are also transposed to their ternary Pd-In$_2$O$_3$-ZrO$_2$ counterpart, where the two ZrO$_2$ phases are intermixed, but typically form separate intergrown particles, as indicated by aberration-corrected scanning electron microscopy (AC-STEM) images, thereby corroborating XRD results (Fig. 4c). In fact, the In$_2$O$_3$ phase is highly dispersed over both ZrO$_2$ polymorphs mostly as monolayer-like structures, which are located on the surface of the highly crystalline support (Supplementary Fig. 11a, b). A similar indium oxide architecture is found for the ternary system prepared by WI (Supplementary Fig. 13b). Interestingly, in all areas studied in the used catalyst, palladium is mostly present as low-nuclearity species that are predominantly associated with indium oxide rather than ZrO$_2$ (Fig. 4d). In contrast, Pd tends to segregate when co-impregnated with indium on *m*-ZrO$_2$

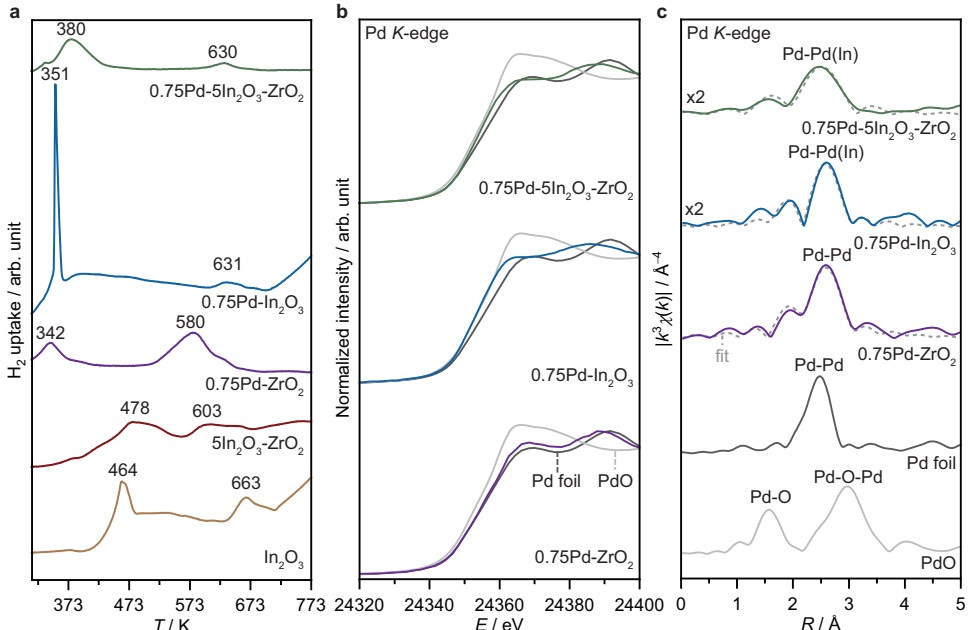

**Fig. 5 | Reducibility and palladium speciation of the Pd-In$_2$O$_3$-ZrO$_2$ catalyst.**
**a** H$_2$-TPR profiles of In$_2$O$_3$-based systems prepared by FSP with Pd-ZrO$_2$ serving as a reference. **b** k-weighted XANES, and (**c**) EXAFS spectra, with fit model and spectra of PdO and metallic Pd serving as references, for catalysts after CO$_2$ hydrogenation for 20 h (Pd K-edge). Activation conditions: $T = 553$ K, $P = 5$ MPa, H$_2$/CO$_2 = 4$, and $GHSV = 48,000$ cm$^3$ h$^{-1}$ g$_{cat}^{-1}$.

and sinters into nanoparticles when supported on pure In$_2$O$_3$ or zirconia by FSP (Supplementary Fig. 12a, b and 13a). This negatively impacts the performance of these systems (Fig. 2) as previously reported[11,20,35,36], and further stresses that the flame preparation enhances synergic effects of combining promoter, active phase, and carrier.

To gather information about catalyst reducibility and palladium speciation, key FSP catalysts were examined by temperature-programmed reduction with hydrogen (H$_2$-TPR) and X-ray absorption spectroscopy at the Pd K-edge (Fig. 5). H$_2$-TPR measurements revealed that surface reduction of In$_2$O$_3$ occurred at much lower temperatures for all Pd-promoted catalysts compared to unpromoted bulk and supported oxide (342–380 K versus 464–478 K, respectively, Fig. 5a), highlighting the assistance of palladium in splitting H$_2$ and forming oxygen vacancies on In$_2$O$_3$. Interestingly, the signal of the ternary Pd-In$_2$O$_3$-ZrO$_2$ catalyst is broader and located at a higher temperature (380 K) than for other palladium-containing samples (Pd-In$_2$O$_3$ and Pd-ZrO$_2$, 351 and 342 K, respectively), hinting at a stronger interaction of Pd and In$_2$O$_3$ on the former and less clustering of the promoter, which is in line with microscopy results (Fig. 4). Analysis of the X-ray absorption near edge structure (XANES, Fig. 5b) confirms that palladium is fully reduced to metallic state on Pd-doped catalysts under reaction conditions, whereas indium remained in cationic form (In$^{3+}$), as also indicated by the XPS results (Supplementary Fig. S14) and in line with previous reports[4,9,10,36,37]. Still, while the XANES spectrum of Pd-ZrO$_2$ resembles that of the reference Pd foil, a white-line shift is evidenced for both Pd-In$_2$O$_3$ and Pd-In$_2$O$_3$-ZrO$_2$, indicating that Pd is most likely bound to indium in these two systems. In principle, Pd-Pd and Pd-In bonds cannot be completely distinguished by fitting the extended X-ray absorption fine structure (EXAFS) spectra (Fig. 5c), owing to the similar scattering factors of these elements. Nonetheless, based on the low amount and high dispersion of Pd as well as the short Pd-Pd(In) bond (2.68 Å) (Supplementary Table 4), it is unlikely that more Pd-Pd than Pd-In bonds are present on the ternary Pd-In$_2$O$_3$-ZrO$_2$ catalyst.

## Kinetic insights on enhanced methanol formation

To shed further light on the origin of the remarkable performance of Pd-In$_2$O$_3$-ZrO$_2$, kinetic analyses were conducted. The apparent activation energies ($E_{app}$) for methanol synthesis and the RWGS reaction (Fig. 6a), obtained from catalytic tests performed at different temperature (Supplementary Fig. 15), revealed that methanol formation is significantly more facile than CO production over both the binary and ternary systems containing In$_2$O$_3$ (67 and 68 kJ mol$^{-1}$ *versus* 100 and 96 kJ mol$^{-1}$, respectively). This rationalizes the high methanol selectivity and yield of these catalysts (Supplementary Fig. 2). This difference in $E_{a,app}$ between the two paths is also observed for Pd-In$_2$O$_3$, but it is much less prominent (*ca.* 12 kJ mol$^{-1}$). The extremely low $E_{a,app}$ of the two reactions over Pd-ZrO$_2$ points towards CO and subsequently MeOH formation through the COOH intermediate on Pd nanoparticles being the predominant reaction mechanism[6,11].

Since palladium is known to facilitate hydrogen activation, apparent reaction orders with respect to this reactant ($n_{H_2}$) were determined from experiments varying the partial pressures of H$_2$ in the feed (Fig. 6b and Supplementary Fig. 16). For both methanol synthesis and the RWGS reactions, $n_{H_2}$ decreased for all systems with respect to the bulk oxide, suggesting that both Pd and ZrO$_2$ increases the abundance of surface H* species that promote the hydrogenation steps, with the promoter exerting the greater effect. H$_2$ splitting forming surface H* is likewise enhanced by larger Pd clusters on Pd/ZrO$_2$, but to a smaller extent than for the other binary systems and particularly the ternary catalyst. Overall, the kinetic data suggest the enhancement of either vacancy creation and/or MeOH formation through the formate pathway over these sites. Still, it is important to emphasize that further studies should be dedicated to acquire detailed insights into the CO$_2$ hydrogenation mechanism on ternary Pd-In$_2$O$_3$/ZrO$_2$ systems, which should also include an experimental investigation using isotopically labelled compounds.

## Oxygen vacancy density by electron paramagnetic resonance spectroscopy

To investigate the formation of oxygen vacancies on Pd-In$_2$O$_3$-ZrO$_2$, electron paramagnetic resonance spectroscopy (EPR) experiments

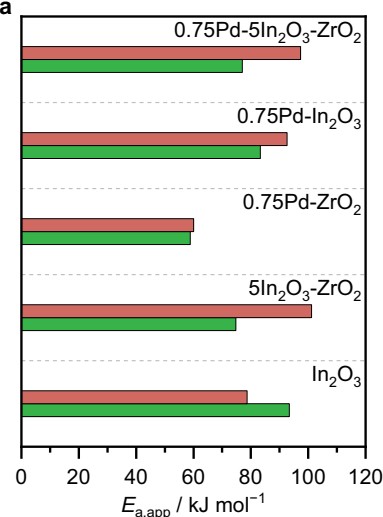

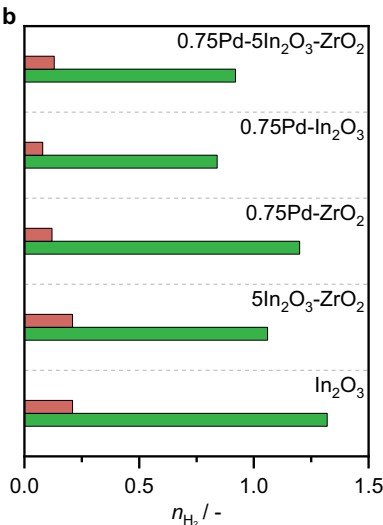

**Fig. 6 | Kinetic analysis of the Pd-In$_2$O$_3$-ZrO$_2$ catalyst in CO$_2$ hydrogenation.** **a** Apparent activation energies ($E_{a,app}$), and (**b**) reaction order with respect to H$_2$ ($n_{H_2}$) for CO$_2$ hydrogenation to methanol (green) and the RWGS reaction (red), over selected catalysts prepared by FSP. Arrhenius plots with data regressions used to determine the activation energy and reaction order values are depicted in Supplementary Figs. 11–12.

were conducted (Fig. 7a, b). When the fresh In$_2$O$_3$/$t$-ZrO$_2$,WI catalyst was analyzed at 20 K, a prominent signal was observed that can be attributed to unpaired electrons trapped in symmetric sites (an approximately isotropic signal at a lower field, $g = 2.004$). Such signals are typically assigned to oxygen vacancies ($V_o$) and commonly referred to as color centers[38]. A very weak signal of this type was observed for FSP In$_2$O$_3$-ZrO$_2$ (Fig. 7a). Another particular EPR feature, which is absent for FSP In$_2$O$_3$-ZrO$_2$ but shared by the In$_2$O$_3$/$t$-ZrO$_2$,WI and Pd/ZrO$_2$ systems in fresh form, is an anisotropic signal with axial $g$ tensor ($g_{xx} = g_{yy} = 1.979$, $g_{zz} = 1.963$) that can be attributed to Zr$^{3+}$ defect sites[38]. Upon reaction, $V_o$ increased considerably, while the Zr$^{3+}$ signal disappeared, which was especially prominent for the In$_2$O$_3$/$t$-ZrO$_2$,WI catalyst. In addition, a new anisotropic signal emerged ($g_{xx} = 2.0025$, $g_{yy} = 2.0036$, $g_{zz} = 2.0043$), which can be assigned to superoxide ions (O$_2^-$) formed on the catalyst surface[38]. In contrast, while the Zr$^{3+}$ signal also vanished for used Pd/ZrO$_2$ materials, no additional features were evidenced. The evolution of $V_o$, Zr$^{3+}$, and O$_2^-$ signals can be explained by the same vacancy-related process. H$_2$ forms new oxygen vacancies ($V_o$) bound to Zr$^{3+}$, the latter then being oxidized to Zr$^{4+}$ by electron

transfer to newly formed $V_o$ that thus also become paramagnetic ($V_o^-$). Similar behavior was also previously reported for pure ZrO$_2$[39]. Finally, the formation of O$_2^-$ on the catalyst surface is likely associated with the release of oxygen from the In$_2$O$_3$ lattice when vacancies are formed.

In principle, the lack of strong vacancy signals over used FSP samples (Fig. 7a), particularly for the ternary Pd-In$_2$O$_3$-ZrO$_2$ system, would suggest that the formation of such sites is not fostered over these catalysts, which is a counterintuitive observation based on their high catalytic performance. While this interpretation is plausible, it is not possible to conclude with certainty based only on EPR measurements performed at low temperatures such as 20 K (Fig. 7a). Specifically, since In$_2$O$_3$ is a semiconductor, a low density of vacancies will indeed produce a sharp and intense signal characteristic of isolated paramagnetic defects, the concentration of which is low enough that they can be considered as non-interacting between each other and with other paramagnetic species, and therefore will be visible at both room and low temperature (20 K, Fig. 7a, d), as previously reported for similar materials[1]. In contrast, an increased density of vacancies will translate into a higher conductivity and in a broad ferromagnetic signal in EPR that is typically better probed at room temperature (Fig. 7b, c, see Supporting Discussion)[40]. Indeed, a very broad signal (*ca.* 5000 G) is present in most of the fresh samples when measured at 303 K (Fig. 7b) and can be attributed to bulk oxygen vacancies in In$_2$O$_3$, which are likely thermally induced in nature and formed due to the harsh temperatures reached in the FSP synthesis. This signal is particularly sensitive to the amount of In$_2$O$_3$ and therefore stronger for the pure active phase. In line with their lower In$_2$O$_3$ content (*ca.* 5 wt.%), fresh In$_2$O$_3$-ZrO$_2$ and Pd-In$_2$O$_3$-ZrO$_2$ showed a weaker signal. Interestingly, compared to other catalysts in used form, a much broader EPR signal is evidenced for these systems, especially for the Pd-In$_2$O$_3$-ZrO$_2$ system. These findings suggest that new oxygen vacancies are formed under reaction conditions, with binary In$_2$O$_3$-ZrO$_2$ and ternary Pd-In$_2$O$_3$-ZrO$_2$ catalysts attaining the highest density of such active sites. In this case, the great concentration of interfacial ZrO$_2$/In$_2$O$_3$ sites associated with the high dispersion of In$_2$O$_3$ over these systems likely promotes a strong delocalization of electrons and ferromagnetism, which are characteristic of high density of vacancies. Furthermore, these results highlight the role of ZrO$_2$, whose interaction with In$_2$O$_3$ is maximized in FSP materials, to foster not only the formation but also stabilization of vacancies, as indicated by the absence of the characteristic signal in used samples of both bulk In$_2$O$_3$ and Pd-In$_2$O$_3$. Lastly, the EPR spectra show the presence of Pd$^{3+}$ in the used ternary catalyst, which is lacking for Pd-In$_2$O$_3$ and Pd-ZrO$_2$, indicating that palladium is strongly interacting with In$_2$O$_3$, likely replacing In$^{3+}$ ions in the structure (Fig. 7b, see Supplementary Discussion), thereby corroborating XANES and microscopy findings.

To further evaluate the formation of oxygen vacancies on the ternary Pd-In$_2$O$_3$-ZrO$_2$ catalyst, an in situ EPR study was performed (Fig. 8a). A continuous and distinct broadening of the vacancy-related signal was observed under flowing H$_2$ with time-on-stream, evidencing that the broad ferromagnetic feature is indeed due to the creation of additional oxygen vacancies, which are augmented by H$_2$ and not just thermally induced. This phenomenon is further illustrated by the trend in peak-to-peak linewidth (Fig. 8b), and translates into the creation of an impurity band in In$_2$O$_3$ (Fig. 8c), which leads to high delocalization of electrons and, consequently, broad ferromagnetic signals in the EPR spectra. In line with these findings, a subsequent switch to flowing CO$_2$ for 2 h resulted in a progressive slight narrowing of the vacancy signal (Fig. 8a, b), which is associated with the newly created sites being partially healed when exposed to an oxidant gas (Fig. 8c). Additionally, since they were not totally annihilated even under a pure CO$_2$ atmosphere, vacancies formed over the ternary Pd-In$_2$O$_3$-ZrO$_2$ catalyst must be particularly stable during the catalytic cycle, which likely contributes markedly to its superior performance.

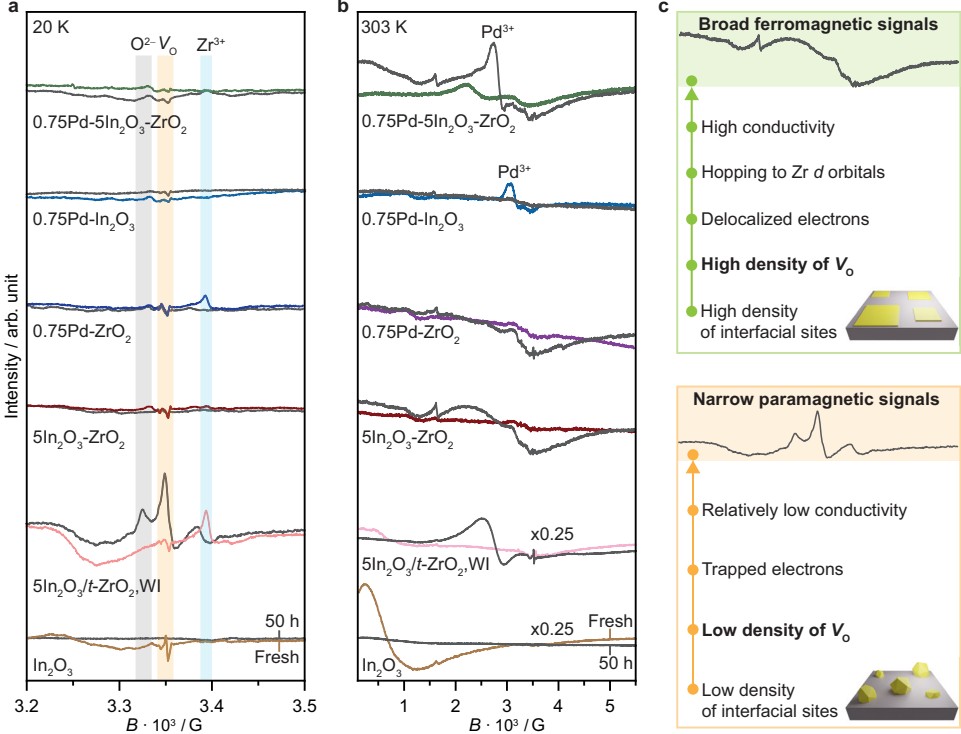

**Fig. 7 | Assessment of oxygen vacancies formation on the Pd-In₂O₃-ZrO₂ catalyst.** Ex situ EPR spectra of In₂O₃-based catalysts prepared by FSP and WI in fresh form and after CO₂ hydrogenation for 50 h measured at (**a**) 20 K and (**b**) 303 K. **c** Scheme describing the nature of EPR signals generated from catalysts displaying distinct densities of interfacial sites and oxygen vacancies. Activation conditions: $T = 553$ K, $P = 5$ MPa, $H_2/CO_2 = 4$, and $GHSV = 48,000$ cm³ h⁻¹ g$_{cat}$⁻¹.

In principle, because a broader ferromagnetic linewidth is associated with a higher density of vacancies, one can expect a correlation between the linewidth of the newly formed signal and the catalytic activity of In₂O₃-ZrO₂-based systems. Interestingly, by normalizing the linewidth values (Supplementary Table 6), we can semi-quantitatively estimate the relative concentration of vacancies ($[V_O]$), which shows a linear dependence with the methanol $STY$, suggesting that $[V_O]$ is a key performance descriptor for Pd-In₂O₃-ZrO₂ catalysts (Fig. 8d). Finally, to provide a comprehensive comparison of ternary Pd-In₂O₃-ZrO₂ catalysts prepared by FSP, WI, and CP under CO₂ hydrogenation conditions, key experimental findings are graphically summarized in Fig. 9. Overall, the superior performance of the Pd-In₂O₃-ZrO₂ attained by FSP is critically related to its ability to ensure a high dispersion and strong interaction between the In₂O₃ and palladium while preventing their incorporation into the bulk of the ZrO₂ carrier, which translates in a higher density of vacancies and greatly favors methanol formation in comparison to other systems prepared by WI and CP.

In summary, the ternary Pd-In₂O₃-ZrO₂ catalyst produced here in a single-step by flame spray pyrolysis (FSP) and featuring excellent atom economy comprises a compelling advance towards the practical implementation of In₂O₃-based systems for methanol production via CO₂ hydrogenation. The use of FSP was pivotal to overcoming synthetic constraints faced by other methods, maximizing the formation and interaction between low-nuclearity palladium species and In₂O₃ platelets and their surface dispersion over the ZrO₂ carrier, which markedly favors methanol formation while suppressing CO production. In contrast to previous reports, a pure *monoclinic*-ZrO₂ phase was not essential because, while more abundant, the particles of the *tetragonal* phase resulting from FSP underwent a surface reconstruction to form a *monoclinic*-like structure upon reaction without inducing sintering of other components. A systematic investigation using ex situ and in situ EPR spectroscopy provided a means to quantify oxygen vacancies, revealing that their density is augmented on the FSP-made Pd-In₂O₃-ZrO₂ system, which show a direct correlation with

its drastically enhanced reactivity, and are most likely stabilized by the ZrO₂ carrier. Overall, the findings highlight the potential of FSP for the design of frontier catalysts in CO₂-based methanol synthesis, and offer a new approach to assess their density of oxygen vacancies. Going beyond this, the use of these quantification and synthesis strategies give prospective for tailoring other relevant reducible oxides acting as heterogeneous catalysts in diverse applications.

## Methods

### Catalyst preparation

Ternary Pd-In₂O₃-ZrO₂ catalysts and reference systems with a nominal metal promoter and indium content described in Supplementary Table 2 were prepared by FSP. Briefly, precursor solutions were prepared by mixing stock solutions of each element in the desired ratio, solvent, and with a total metal concentration of *ca.* 0.5 mol kg⁻¹$_{solution}$ (see Supplementary Table 1). Thereafter, the precursor solutions were pumped through a 0.4 mm nozzle at a flow rate of 5 cm³ min⁻¹ and dispersed into a fine spray by flowing oxygen at 1.5 bar at a flow rate of 5 dm³ min⁻¹. The spray was ignited by a supporting flame generated using 2.4 and 1.2 dm³ min⁻¹ of oxygen and methane, respectively, with average temperatures of 2500−3000 K and very fast cooling rates (*ca.* 10⁶ K s⁻¹)[30]. The resulting nanoparticles were collected on a glass fiber filter (GF/A-6) and used in CO₂ hydrogenation without further treatment. The FSP setup is described in detail elsewhere[41]. The syntheses of all other materials reported in this study are detailed in the Supplementary Methods.

### Catalyst characterization

Inductively coupled plasma optical emission spectroscopy (ICP-OES) was conducted using a Horiba Ultra 2 instrument equipped with a photomultiplier tube detector. Samples were digested with the aid of microwave irradiation using a mixture of HCl (Alfa Aesar, 36 wt.%), H₂SO₄ (Alfa Aesar, 95 wt.%), and HF (Sigma Aldrich, 48 wt.%) with a volume ratio of 2:1:0.5, followed by neutralization with a saturated

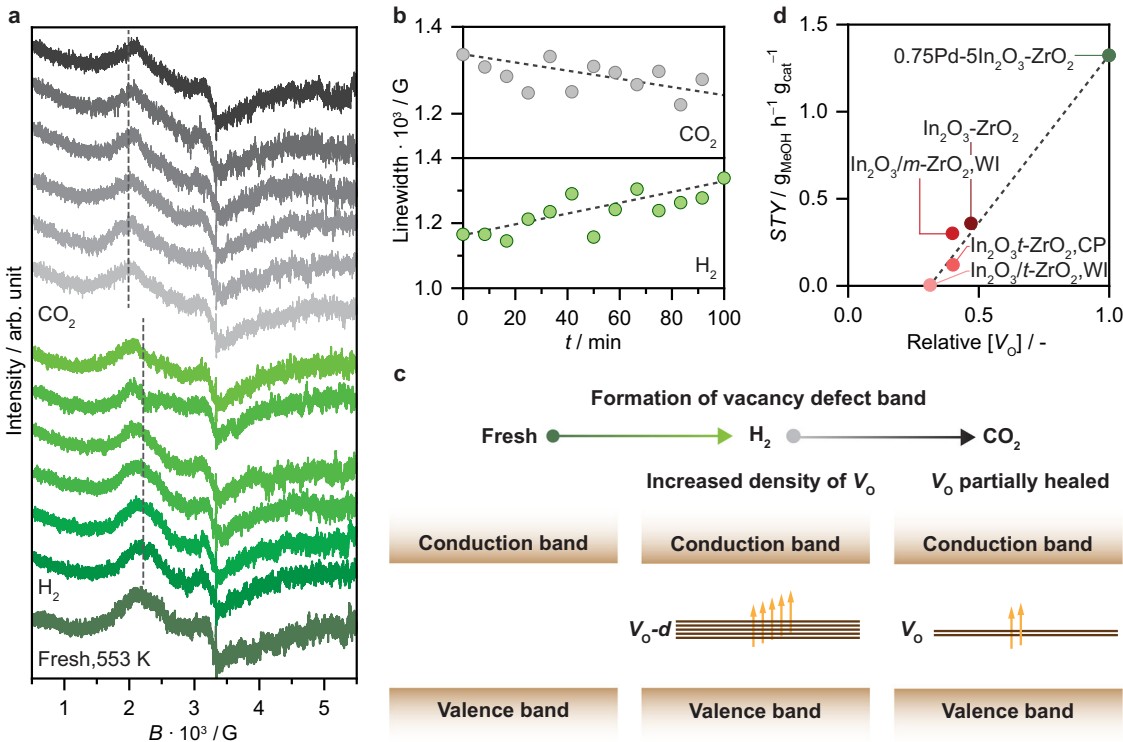

**Fig. 8 | Oxygen vacancy dynamics and performance descriptor for Pd-In$_2$O$_3$-ZrO$_2$ catalysts in CO$_2$ hydrogenation. a** In situ EPR spectra of the 0.75Pd-5In$_2$O$_3$-ZrO$_2$ catalyst prepared by FSP measured first under flowing H$_2$ (100 min) and then CO$_2$ (100 min). **b** Peak-to-peak linewidth values over time derived from EPR spectra shown in **a**. **c** Band diagrams describing the impact of oxygen vacancies on the electronic structure of In$_2$O$_3$ upon treatment in H$_2$ and CO$_2$. Activation conditions: $T = 553$ K, $P = 0.1$ MPa, $m_{cat} = 10$ mg, and flow(H$_2$) = flow(CO$_2$) = 20 cm$^3$ min$^{-1}$.

**d** Methanol space-time yield (*STY*) over the 0.75Pd-5In$_2$O$_3$-ZrO$_2$ catalyst prepared by FSP and reference materials during CO$_2$ hydrogenation (Fig. 2) as a function of the relative concentration of oxygen vacancies ([$V_O$]), which was obtained by normalizing the peak-to-peak linewidth of the newly formed signal (Supplementary Table 6). Activation conditions: $T = 553$ K, $P = 5$ MPa, H$_2$/CO$_2$ = 4, *GHSV* = 48,000 cm$^3$ h$^{-1}$ g$_{cat}$$^{-1}$, and *TOS* = 50 h.

solution of HBO$_3$ (Fluka, 99%). X-ray fluorescence spectroscopy (XRF) was performed using an Orbis Micro-EDXRF spectrometer equipped with a Rh source operated at 35 kV and 500 µA and a silicon drift detector. Nitrogen sorption at 77 K was carried out using a Micromeritics TriStar II analyzer. Prior to the measurements, samples were degassed at 473 K under vacuum for 12 h. The total surface area ($S_{BET}$) was determined using the Brunauer-Emmet-Teller (BET) method. X-ray diffraction (XRD) was conducted using a Rigaku SmartLab diffractometer with a D/teX Ultra 250 detector using Cu K$\alpha$ radiation ($\lambda = 0.1541$ nm) and operating in the Bragg-Brentano geometry. Data was acquired in the 20–70° 2$\theta$ range with an angular step size of 0.025° and a counting time of 1.5 s per step. High-angle annular dark-field scanning transmission electron microscopy (HAADF-STEM) images and energy-dispersive X-ray spectroscopy maps were collected using a Talos F200X instrument operated at an acceleration potential of 200 kV. Aberration-corrected AC-STEM images were acquired on a double-corrected JEM-ARM300F microscope (GrandARM, JEOL), equipped with a dual energy dispersive X-ray (EDX) spectroscopy system, and operated at 300 kV in Z-contrast mode. Samples were dusted on lacey-carbon coated copper or nickel grids. Temperature-programmed reduction with H$_2$ (H$_2$-TPR) was conducted at ambient pressure using a Micromeritics AutoChem HP II equipped with a thermoconductivity detector. Samples were loaded in a stainless-steel tube, dried at 423 K in a Ar flow for 1 h, and cooled down to 313 K (20 K min$^{-1}$). The temperature-programmed reduction was then carried out using 5 vol% H$_2$/Ar and increasing the temperature to 973 K (10 K min$^{-1}$). Raman spectroscopy was performed using a Horiba: LabRAM HR Evolution UV-VIS-NIR confocal Raman system comprising a 325 nm HeCd laser with 2.5 mW power, a 40 × objective lens with a numerical aperture of 0.95 (Nikon PlanApo), and a fiber coupled

grating spectrometer (1800 lines per mm). Spectra were collected in a single run with 60 s acquisition time. For data evaluation, the intensities were normalized by the highest measured intensity after subtraction of a linear background. X-ray photoelectron spectroscopy (XPS) was performed using a Physical Electronics (PHI) Quantum 2000 X-ray photoelectron spectrometer featuring monochromatic Al K$\alpha$ radiation, generated from an electron beam operated at 15 kV and 32.3 W, and a hemispherical capacitor electron-energy analyzer, equipped with a channel plate and a position-sensitive detector. Samples were firmly pressed onto aluminum foil patches, which were then mounted onto a sample platen and introduced into the spectrometer. Analyses were conducted under ultra-high vacuum (residual pressure = $6 \times 10^{-9}$ Pa) with an electron take-off angle of 45°, operating the analyzer in the constant pass energy mode. X-ray absorption spectroscopy (XAS) was measured at the SuperXAS beamline at the Swiss Light Source[42]. The incident beam was provided by the 2.9 T super bent source. The X-ray beam was collimated by Pt-coated (> 20 keV), mirror at 2.9 mrad and focused by a toroidal Pt-coated mirror. The energy was selected by a Si(111) channel-cut monochromator[43], and calibrated using a Pd foil (24.350 keV), which was measured simultaneously with the specimen of interest. The incident X-ray beam was focused on a 0.25 × 1 mm$^2$ spot. Activated samples ($T = 553$ K, $P = 5$ MPa, H$_2$/CO$_2$ = 4, *GHSV* = 48,000 cm$^3$ h$^{-1}$ g$_{cat}$$^{-1}$, and TOS = 20 h) were transferred from the reactor to a quartz capillary and sealed under inert atmosphere. Reference PdO was measured in transmission mode as a pellet diluted with cellulose. Three 15 cm long ionization chambers filled with 50 vol.% N$_2$ in Ar at 2 bar were used to monitor the incident beam intensity, transmission through the sample, and the reference foil. The XAS spectra of 0.75Pd-In$_2$O$_3$ sample was measured using a quick fluorescence detection mode with a PIPS diode

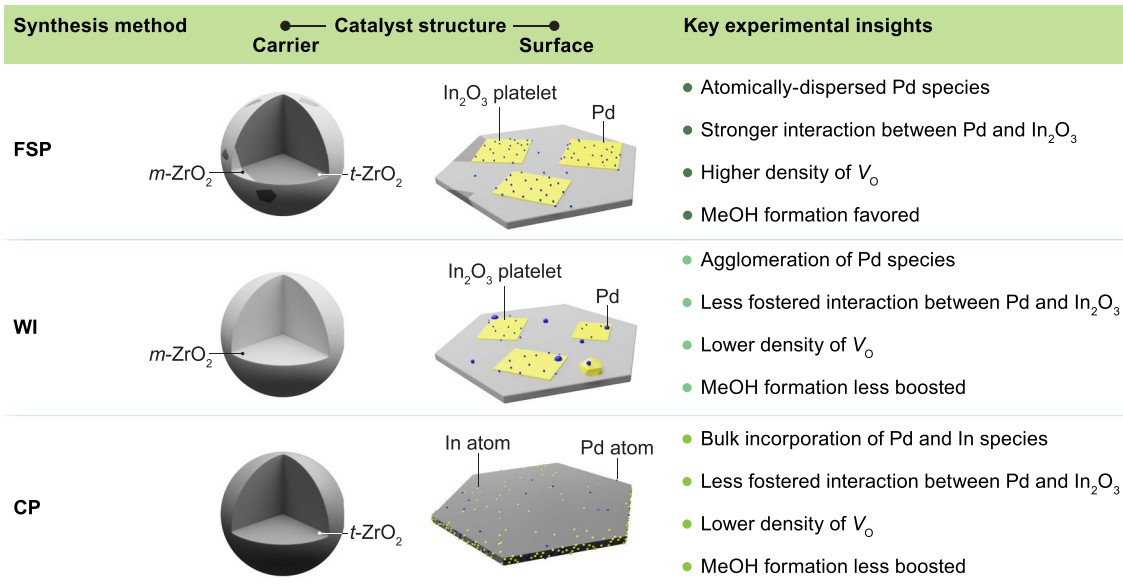

**Fig. 9 | Comparison of ternary Pd-In$_2$O$_3$-ZrO$_2$ catalysts attained by distinct synthesis methods.** Summary of key geometric and electronic features dictating performance of ternary Pd-In$_2$O$_3$-ZrO$_2$ catalysts.

detector (1 Hz monochromator oscillation, 5–10 min data acquisition). The XAS spectra of 0.75Pd-ZrO$_2$ and 0.75Pd-In$_2$O$_3$-ZrO$_2$ were measured in fluorescent mode using a 5-element SDD detector (SGX) and step-by-step data acquisition mode. The spectra were calibrated and averaged with the in-house developed ProEXAFS software and further analyzed using the Demeter software package[44]. $k^3$-weighted extended X-ray absorption fine structure (EXAFS) spectra were fitted in the optimal $k$- and $R$-windows (Supplementary Table 4). An amplitude reduction factor ($S_0^2$) of 0.82 was determined by fitting of the EXAFS spectrum of a Pd foil. The scattering paths for the fitting were produced using known crystallographic structures of metallic Pd, *tetragonal* PdO, and cubic In$_2$O$_3$. Continuous wave (CW) electron paramagnetic resonance (EPR) spectroscopy experiments were performed on a Bruker Elexsys E500 spectrometer equipped with an Oxford helium (ESR900) cryostat operating at X-band frequencies using a ER4122SHQE Bruker EPR Resonator. All ex situ CW EPR spectra were acquired at 20 K and room temperature using spectrometer settings described in Supplementary Table 5. All measured $g$-factors were offset-corrected against a known standard (free radical 1,1-diphenyl-2-picrylhydrazyl). In situ EPR spectroscopy experiments were performed using a custom-built setup and according to spectrometer settings summarized in Supplementary Table 5. A quartz capillary ($d_i = 0.8$ mm) was loaded with the catalyst, and placed inside an EPR quartz tube (Wilmad; $d_i = 2.8$ mm). The EPR tube was placed at the center of a homemade water-cooled high-temperature resonator[45], which was installed into a CW-EPR spectrometer (Bruker EMX) operating at X-band frequencies. The gas flow was directed from the top of the capillary through the catalyst bed and then through the space between the capillary outer walls and the EPR tube inner walls. The reactor was heated in an Ar flow to the target temperature ($T = 553$ K) and allowed to stabilize for 20 min. The two reactant gases were sequentially admitted to the reactor, i.e., a H$_2$ (20 cm$^3_{STP}$ min$^{-1}$) was flown for 2 h, followed by CO$_2$ (20 cm$^3_{STP}$ min$^{-1}$) for 2 h. The gases were dosed by a set of digital mass flow controllers and the outcome was monitored on-line via a Pfeiffer Vacuum Thermo-Star GSD 320 T1 mass spectrometer. The EPR spectra were continuously acquired upon flowing the gases and separately stored, using a 2D acquisition mode, thus enabling a time-resolved monitoring of the process.

## Catalyst evaluation

The gas-phase hydrogenation of CO$_2$ to methanol was performed in a PID Eng&Tech high-pressure continuous-flow setup comprising four parallel fixed-bed reactors, as described elsewhere[7]. Undiluted catalysts (mass, $m_{cat} = 0.1$ g; particle size = 0.2–0.4 mm) were loaded into each reactor tube (internal diameter 4 mm), held in place by a quartz-wool bed set on a quartz frit, and purged in flowing He (40 cm$^3$ STP min$^{-1}$, PanGas, 4.6) for 30 min at ambient pressure. Under the same flow, the pressure was increased to 5.5 MPa for a leak test. The reaction was carried out by feeding a mixture of H$_2$ (PanGas, 5.0), CO$_2$ (40 vol.% in H$_2$, Messer, 4.5), with a molar H$_2$/CO$_2$ ratio of 4 at 553 K, 5 MPa, and a gas hourly space velocity (GHSV) of 48,000 cm$^3$ $_{STP}$ h$^{-1}$ g$_{cat}^{-1}$, unless stated otherwise. The selectivity of the catalysts was compared at a constant degree of CO$_2$ conversion ($X_{CO_2}$) of *ca.* 3% by adjusting the GHSV for each system using catalysts of variable masses diluted in quartz sand. To determine activation energies of methanol and CO formation, the reaction was initiated at 473 K and increased stepwise to 573 K in increments of 10 K at 5 MPa ($m_{cat} = 0.05$ g, and H$_2$/CO$_2$ = 4 at specified GHSV). Reaction orders with respect to H$_2$ were acquired by applying a constant flow of CO$_2$ (8 cm$^3$ STP min$^{-1}$) and increasing the flow of H$_2$, while using He to balance the total flow to 44 cm$^3$ STP min$^{-1}$ for a GHSV of 48,000 cm$^3$ $_{STP}$ h$^{-1}$ g$_{cat}^{-1}$ ($m_{cat} = 0.055$ g) at 553 K and 5 MPa.

The effluent streams were analyzed by gas chromatography every 1 h. Response factors ($F_i$) for each compound $i$, respective to the internal standard (20 vol.% C$_2$H$_6$ in He, Messer, purity 3.5), in the GC analysis were determined by Eq. 1:

$$F_i = \frac{A_{C_2H_6}/n_{C_2H_6}^{in}}{A_i/n_i^{in}} \tag{1}$$

where $A_i$ is the integrated area determined for the peak of compound $i$ and $n_{in}$ is the corresponding known molar flow at the reactor inlet. An average of 5 points around the expected analyte concentration was used. The unknown effluent molar flow of compound $i$ was determined using Eq. 2:

$$\dot{n}_i^{out} = \frac{A_i \times F_i}{A_{C_2H_6}} \times \dot{n}_{C_2H_6}^{out} \tag{2}$$

Conversion ($X_i$), selectivity ($S_i$), and production rate ($r_i$) were calculated using Eqs. 3–5:

$$X_i = \frac{\dot{n}_i^{in} - \dot{n}_i^{out}}{\dot{n}_i^{in}} \qquad (3)$$

$$S_i = \frac{\dot{n}_i^{in} - \dot{n}_i^{out}}{\dot{n}_{CO_2}^{in} - \dot{n}_{CO_2}^{out}} \qquad (4)$$

$$r_i = \frac{\dot{n}_i^{in} - \dot{n}_i^{out}}{m_{cat}} \qquad (5)$$

The methanol space-time yield (STY) is the product of $r_{MeOH}$ and the molar weight of methanol (32.04 g mol$^{-1}$). The carbon balance was determined for each experiment according to Eq. 6:

$$\varepsilon_C = \left(1 - \frac{\dot{n}_{CO_2}^{out} + \dot{n}_{MeOH}^{out} + \dot{n}_{CO}^{out}}{\dot{n}_{CO_2}^{in}}\right) \qquad (6)$$

and was always within a 5% margin.

## Data availability

Data presented in the main figures of the manuscript are publicly available through the Zenodo repository (https://doi.org/10.5281/zenodo.6511235)[46]. Further data supporting the findings of this study are available in the Supplementary Information. All other relevant source data are available from the corresponding author upon request. Source data from the main figures are provided with this paper. Source data are provided with this paper.

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

## Acknowledgements

This publication was created as part of NCCR Catalysis (grant number 180544), a National Centre of Competence in Research funded by the Swiss National Science Foundation. The Scientific Center for Optical and Electron Microscopy (ScopeM) at the ETH Zurich and the SuperXAS beamline at PSI, are thanked for access to their facilities. Dr. Frank Krumeich, Dr. Milivoj Plodinec, and Mr. Dario Faust Akl are thanked for acquiring the HAADF-STEM-EDX and AC-STEM-EDX data. We are grateful to Prof. Christophe Copéret, Mr. Enzo Brack, and Mr. Jan Alfke for assistance with sample preparation for XAS measurements. T.Z. thanks the Agency for Science, Technology and Research (A*STAR) Singapore for support through a graduate fellowship.

## Author contributions

J.P.-R. conceived and coordinated the study. T.P.A., C.M., S.M., and J.P.-R. wrote the article with input from all other co-authors. T.P.A. synthesized the catalysts, contributed to their characterization, and conducted the catalytic tests. M.A. and G.J. conducted the electron paramagnetic resonance spectroscopy studies. P.O.W., K.M.E., R.N.G., and W.J.S. prepared the catalysts using the flame spray pyrolysis method. T.Z. conducted kinetic analyses. S.M. coordinated acquisition and performed the evaluation of electron microscopy analyses. O.V.S. supervised acquisition and evaluation of X-ray absorption spectroscopy data. All authors contributed to the writing of the manuscript.

## Competing interests

The authors declare no competing interests.
