## [Peer Review File · Nature Communications]

Title: Flame-made ternary Pd-In₂O₃-ZrO₂ catalyst with enhanced oxygen vacancy generation for CO₂ hydrogenation to methanolREVIEWER COMMENTS

Reviewer #1 (Remarks to the Author):

Unique architecture of ternary Pd-In₂O₃-ZrO₂ catalyst prepared by flame spray pyrolysis unlocks high performance in CO₂ hydrogenation to methanol

This work reported the Pd-In₂O₃-ZrO₂ catalyst prepared by a FSP method, which showed the high activity to the hydrogenation of CO₂ to methanol, being 3-5 times higher than those catalysts from traditional approaches. The structure of catalysts has been characterized using multi techniques, such as TEM, XRD and in situ EPR etc.. However, after reviewing the advances in the presented catalyst, my impression is that the work did not make a significant progress in this topic. The following suggestions may help for consideration.

(1) Up to now, the conversion of CO₂ is still less than 10% in single pass, far below industrial criterion. Since the activity, or to say, STY, is too low. With this respect, the way the research team working on is in right track. However, the selectivity declined mildly, below 90%.

(2) According to results SI Figure 1, no matter how the components are, with or without Pd, ZrO₂, the selectivity to methanol remained unchanged. It is well known that the structure InO_x change little during the reaction. This paper gave less information for the elucidating this phenomenon.

(3) As shown in SI Figure 1, the high activity for FSP Pd-In-Zr has been attributed to more oxygen vacancy creating on ZrO₂ surface. If comparing the performance of sample 8 (CP, Pd-In) and sample 10 (FSP, Pd-In-Zr), they showed a very close performance, ZrO₂ seems contributes little to the improvement of catalysts. Thus, all studies became groundless.

(4) Another reason for this high activity of FSP- Pd-In-Zr is assumed to quick H₂ dissociation, it may not be true, because H₂ dissociation is very fast in the surface with abundant of oxygen vacancy.

(5) The mechanism for CO₂ activation on the surface of those catalysts may be worth to study in the further.

Reviewer #2 (Remarks to the Author):

In this manuscript, the authors have investigated a flame spray pyrolysis (FSP) method to prepare ternary Pd-In₂O₃-ZrO₂ catalyst prepared in one step manner. The catalyst prepared via FSP route shows high methanol productivity from hydrogenation of CO₂ and the observed activity is attributed to an enriched density of oxygen vacancies on the surface of catalyst and high dispersity of active components. Generally speaking, the catalysts derived from different preparative routes (FSP, CP & WI) were studied with comprehensive analytical characterization. I must say that the work was executed with care and the paper presentation format is excellent. The detailed investigation would then be very useful the researchers in the field of carbon dioxide utilization. Thus, the paper can be recommended for its publication in this Journal after the following revisions:

(a) MeOH productivity (such as the statement on page 4: “which reaches a record productivity of 1.3 gMeOH h⁻¹gcat⁻¹”) also depends on reactant flow rate adopted, or gas hourly space velocity (GHSV). Therefore, a comparison table for the current work and literature reports under the same GHSV should be presented in Supplementary Information.

(b) Estimation of the thickness of surface In₂O₃ layer should be done for both FSP and WI catalyst samples.

(c) While single-atom In is detected in Supplementary Figure 10b, how to confirm they are indeed In adatoms instead of Pd adatoms?

(d) A comparison of surface texture (i.e., surface area and porosity) and morphology of different particles prepared by different methods (FSP, CP and WI types). This piece of information is important as it determines the catalyst performance in general. Three catalysts must have similar (if not identical) surface area and porosity in order to have a fair comparison on other parameters that dictate the catalytic performance.

(e) Similar structural analysis (i.e., Supplementary Figure 10) should be done for those prepared by WI method in order to demonstrate the catalyst model described in Figure 9 (WI-derived catalyst), that is, forming the surface sheets of In₂O₃.

Manuscript NCOMMS-22-21949-T - Response to Reviewers

Notes: Comments in *blue* - Replies in black - Actions in **bold**

Indicated page, line, figure, or reference numbers refer to the revised manuscript and/or supporting information with changes highlighted

Reviewer #1

Unique architecture of ternary Pd-In₂O₃-ZrO₂ catalyst prepared by flame spray pyrolysis unlocks high performance in CO₂ hydrogenation to methanol. This work reported the Pd-In₂O₃-ZrO₂ catalyst prepared by a FSP method, which showed the high activity to the hydrogenation of CO₂ to methanol, being 3-5 times higher than those catalysts from traditional approaches. The structure of catalysts has been characterized using multi techniques, such as TEM, XRD and in situ EPR etc. However, after reviewing the advances in the presented catalyst, my impression is that the work did not make a significant progress in this topic. The following suggestions may help for consideration.

We thank the Reviewer for assessing our study. Still, we regret that they did not appreciate the significance of our contribution, which is well recognized by Reviewer #2. We believe that our contribution brings unique progress to the field of CO₂ hydrogenation because:

- Despite being considered a pivotal strategy for realizing a prospective industrial implementation of In₂O₃-based catalysts, no work thus far has attempted to integrate palladium, zirconia, and indium oxide into a ternary Pd-In₂O₃/ZrO₂ system. This task is indeed very challenging as the effect of simply combining the three components does not necessarily translate into a linear performance enhancement.
- We have not only pioneered the development of ternary Pd-In₂O₃/ZrO₂ catalysts but, more importantly, identified flame spray pyrolysis (FSP) as a key synthesis method to effectively produce materials with a unique architecture, which, as pointed out by the Reviewer, display up to 3-5 times higher activity than that of counterpart catalysts prepared by traditional synthetic approaches, such as co-precipitation and wet impregnation.
- We reshape the understanding of the effect of ZrO₂ polymorphs on In₂O₃-catalyzed methanol synthesis, showcasing that a pure *monoclinic* phase is not critical to unlocking high performing catalytic systems.
- Last but not least, we address the long-standing challenge of characterizing and quantifying oxygen vacancies on In₂O₃-based catalysts, thereby advancing understanding of this critical performance descriptor and offering a new approach that can be potentially extended to assess the density of oxygen vacancies on other relevant reducible oxides acting as heterogeneous catalysts for diverse applications.

To better highlight these key aspects, which may not have been recognized in our original contribution, **we have amended the title and rewritten the abstract, introduction, and conclusions.**

1. Up to now, the conversion of CO₂ is still less than 10% in single pass, far below industrial criterion. Since the activity, or to say, STY, is too low. With this respect, the way the research team working on is in right track. However, the selectivity declined mildly, below 90%.

We strongly disagree with the validity of this comment. CO₂ hydrogenation to methanol is an equilibrium limited transformation. As such, thermodynamics dictates that the maximum single-pass CO₂ conversion (X_{CO_2}) reachable by an ideal catalyst would be ca. 30% under the reaction conditions applied in our study ($T = 553$ K, $P = 5$ MPa, and $H_2/CO_2 = 4$).^[1] Additionally, it is important to emphasize that even at the same temperature, pressure, and H_2/CO_2 ratio, X_{CO_2} is also significantly impacted by the gas-hourly space velocity ($GHSV$), with its absolute value decreasing as $GHSV$ increases.^[2] Therefore, based on the aforementioned discussion and considering the high $GHSV$ used in our study to investigate catalysts

in the kinetic regime and thus assess their intrinsic performance ($48,000 \text{ cm}^3 \text{ h}^{-1} \text{ g}_{\text{cat}}^{-1}$, ca. 2-fold higher than that generally applied in other studies), the X_{CO_2} of our Pd-In₂O₃/ZrO₂,FSP catalyst, which is actually ca. 12%, is certainly not far from what is expected from industrial standards. To further clarify this point, **we have added the Supplementary Table 3 comparing X_{CO_2} , methanol selectivity (S_{MeOH}), and methanol space-time yield (STY) at similar reaction conditions of various relevant catalysts in CO₂ hydrogenation to methanol.** As shown in **Supplementary Table 3**, the STY of our ternary system is 1-2 orders of magnitude higher than existing systems in relation to the indium content, strengthening the potential implementation of this process. In addition, since S_{MeOH} highly depends on the X_{CO_2} levels at which it is measured, with the former generally decreasing as the latter increases,^[3] attaining a S_{MeOH} of ca. 90% at a high X_{CO_2} level such as 12% is truly outstanding and confirms the intrinsic high selectivity of our catalyst. [1] *Ind. Eng. Chem. Res.* **2018**, 57, 4081–4094. [2] *Chem. Rev.* **2020**, 120, 7984–8034. [3] *Chem. Soc. Rev.*, **2020**, 49, 1385-1413.

2. According to results SI Figure 1, no matter how the components are, with or without Pd, ZrO₂, the selectivity to methanol remained unchanged. It is well known that the structure InO_x change little during the reaction. This paper gave less information for the elucidating this phenomenon.

S_{MeOH} data displayed in **Supplementary Figure 1** was measured at distinct X_{CO_2} levels, therefore, it cannot be unequivocally used to compare S_{MeOH} among different catalysts. However, **Supplementary Figure 2** depicts S_{MeOH} measured at the same X_{CO_2} level for all catalysts prepared by FSP, which in contrast to the Reviewers' claim, confirms that the presence of palladium and ZrO₂ improves S_{MeOH} when compared to that of bulk In₂O₃. Similar observations were also reported for binary Pd-In₂O₃ and In₂O₃/*m*-ZrO₂ systems synthesized by co-precipitation and wet-impregnation, respectively.^[1] Concerning the In₂O₃ structure, it is well-known that it can significantly change upon reaction, depending on the synthesis method, the type of carrier and metal promoter, and reaction conditions used.^[2] In fact, we show an example in our study in which In₂O₃ sinters into large particles upon reaction when supported on *tetragonal* zirconia by wet impregnation, whereas it remains well-dispersed as monolayer-like platelets when deposited on the corresponding *monoclinic* polymorph (**Figure 4a**). Additionally, there have also been reports showing that over-reduction of bulk In₂O₃ into inactive metallic indium can take place during reaction owing to the presence of carbon monoxide generated as byproduct and metal promoters.^[3] [1] *Nat. Commun.* **2019**, 10, 3377; *ACS Catal.* **2020**, 10, 1133. [2] *ACS Catal.* **2021**, 11, 1406; *ACS Catal.* **2020**, 10, 10060; *Nat. Commun.* **2019**, 10, 3377. [3] *J. Am. Chem. Soc.* **2019**, 141, 13497; *Nat. Commun.* **2019**, 10, 3377; *Adv. Energy Mater.* **2022**, 12, 2103707.

3. As shown in SI Figure 1, the high activity for FSP Pd-In-Zr has been attributed to more oxygen vacancy creating on ZrO₂ surface. If comparing the performance of sample 8 (CP, Pd-In) and sample 10 (FSP, Pd-In-Zr), they showed a very close performance, ZrO₂ seems contributes little to the improvement of catalysts. Thus, all studies became groundless.

As mentioned in the response to **Comment #2**, S_{MeOH} data displayed in **Supplementary Figure 1** was measured at distinct X_{CO_2} levels, therefore, it cannot be unequivocally used to compare S_{MeOH} among different catalysts. We attributed the high performance of Pd-In₂O₃/ZrO₂,FSP catalysts to augmented density of oxygen vacancies created on In₂O₃, which is fostered by both Pd and ZrO₂, as uncovered by electron paramagnetic resonance (EPR) spectroscopy (**Figure 7 and 8**). Additionally, while the Reviewer claims that no significant difference in performance exists between the Pd-In₂O₃,CP and Pd-In₂O₃/ZrO₂,FSP catalysts, if we compare their methanol productivity (see STY in **Figure 2** and **Supplementary Table 3**), we observe that methanol productivity over the ternary catalyst is 25% and 97% superior to that of Pd-In₂O₃,CP, on the basis of mass of catalyst and indium, respectively. Therefore, it seems clear that ZrO₂ greatly contributes to improve performance, particularly because it does so while requiring a significantly lower content of indium, which alike palladium is also a scarce and expensive material. **This point has been further highlighted on page 9, lines 188-190.**

4. Another reason for this high activity of FSP- Pd-In-Zr is assumed to quick H₂ dissociation, it may not be true, because H₂ dissociation is very fast in the surface with abundant of oxygen vacancy.

Thank you for raising this important point. Indeed, H₂ dissociation is considered virtually barrierless on Pd-In₂O₃/ZrO₂,FSP, as suggested by measurements of apparent reaction order in H₂ (**Figure 6b**). Still, this most likely does not occur exclusively because of an abundant density of oxygen vacancies, but rather due to a high concentration of active ensembles comprising both isolated palladium atoms and oxygen vacancies, as previously reported for metal promoted-In₂O₃ catalysts.^[1] In fact, it is well-documented that H₂ activation is energetically demanding and limits methanol synthesis over unpromoted In₂O₃, since oxygen vacancies split H₂ in a heterolytic manner while simultaneously activating CO₂.^[2] For this reason, deposition on a *monoclinic* zirconia carrier and introducing a palladium promoter have been used as efficient strategies to improve the performance of bulk In₂O₃.^[3] While both palladium and *m*-ZrO₂ improve H₂ splitting by creating additional vacancies on In₂O₃, the former further contributes to a greater extent to this process, since palladium itself activates H₂ in a homolytic manner, which is barrierless compared to heterolytic H₂ activation on oxygen vacancies and suggested to increase the abundance of surface H* species that promote the hydrogenation steps. Overall, we agree with the Reviewer and hypothesize that the high performance of Pd-In₂O₃/ZrO₂,FSP most likely stems from its ability to facilitate specific C-H hydrogenation steps rather than H₂ dissociation itself. **We have amended this discussion in the revised manuscript on page 15, lines 335-337.** [1] *Nat. Commun.* **2019**, 10, 3377; *Adv. Energy Mater.* **2022**, 12, 2103707. [2] *J. Catal.* **2018**, 361, 313; *ACS Catal.* **2021**, 11, 1406; *Nat. Commun.* **2019**, 10, 3377. [3] *Nat. Commun.* **2019**, 10, 3377; *ACS Catal.* **2021**, 11, 1406.

5. The mechanism for CO₂ activation on the surface of those catalysts may be worth to study in the further.

We thank the Reviewer for this valuable suggestion. Indeed, shedding light on the mechanism for CO₂ activation on ternary Pd-In₂O₃/ZrO₂ systems is worth studying further. Still, it would require a detailed experimental investigation, comparing both ternary systems and their binary counterparts through operando methods using isotopically labeled compounds, and therefore deserves a dedicated study. **We have added a sentence to the amended manuscript pointing to the relevance of such a study in the future (page 15, lines 341-343).**

Reviewer #2

In this manuscript, the authors have investigated a flame spray pyrolysis (FSP) method to prepare ternary Pd-In₂O₃-ZrO₂ catalyst prepared in one step manner. The catalyst prepared via FSP route shows high methanol productivity from hydrogenation of CO₂ and the observed activity is attributed to an enriched density of oxygen vacancies on the surface of catalyst and high dispersity of active components. Generally speaking, the catalysts derived from different preparative routes (FSP, CP & WI) were studied with comprehensive analytical characterization. I must say that the work was executed with care and the paper presentation format is excellent. The detailed investigation would then be very useful the researchers in the field of carbon dioxide utilization. Thus, the paper can be recommended for its publication in this Journal after the following revisions:

We are delighted to read the Reviewer's positive feedback regarding the impact of our study. We have carefully acted upon all their suggestions, which were highly valuable.

1. MeOH productivity (such as the statement on page 4: "which reaches a record productivity of 1.3 g_{MeOH} h⁻¹ g_{cat}⁻¹") also depends on reactant flow rate adopted, or gas hourly space velocity (GHSV). Therefore, a comparison table for the current work and literature reports under the same GHSV should be presented in Supplementary Information.

We agree with the Reviewer on this very relevant point. Accordingly, **we have added Supplementary Table 3 comparing X_{CO₂}, S_{MeOH}, and STY at similar reaction conditions of various relevant catalysts in CO₂ hydrogenation to methanol**, which is included in the response to **Comment #1 of Reviewer #1**. Additionally, **we have also replaced the word "record" by "outstanding"** as the former can have an ambiguous meaning.

2. Estimation of the thickness of surface In₂O₃ layer should be done for both FSP and WI catalyst samples.

Based on the aberration-corrected scanning transmission electron microscopy (AC-STEM) analysis (e.g., as presented in **Supplementary Figures 11 and 13**), the In₂O₃ phase in the catalyst prepared by both flame spray pyrolysis and wet impregnation appears to wet the surface strongly, leading to a typical thickness of no more than a monolayer thick. This observation agrees with the high and uniform dispersion of indium over the carrier particles evidenced in elemental maps of the catalyst acquired by energy dispersive X-ray (EDX) spectroscopy (**Figure 4d**). It is also consistent with structures we previously observed for In₂O₃ supported on monoclinic zirconia including single and groups of indium oxide adatoms.^[1] **We have discussed the thickness of the In₂O₃ layers in the caption of Supplementary Figure 11.** [1] *ACS Catal.* **2020**, 10, 1133.

3. While single-atom In is detected in Supplementary Figure 10b, how to confirm they are indeed In adatoms instead of Pd adatoms?

The Reviewer correctly points out that that it is not possible to differentiate between palladium and indium atoms based solely on AC-STEM images. Our assignment takes into account several other factors i) EDX maps indicate a higher dispersion of In on the ZrO₂ carrier than of Pd (**Figure 4d**), ii) as mentioned above isolated adatoms of indium have been previously observed for binary systems comprising In₂O₃ supported on *m*-ZrO₂,^[1] and iii) palladium species supported on ZrO₂ by flame spray pyrolysis show a greater tendency to form nanoparticles upon reaction than when the metal is supported on In₂O₃, confirming the less preferential stabilization on ZrO₂ (**Supplementary Figure 12**). **We have elaborated on the discussion of the assignment of In adatoms in the caption of Supplementary Figure 11.** [1] *ACS Catal.* **2020**, 10, 1133.

4. A comparison of surface texture (i.e., surface area and porosity) and morphology of different particles prepared by different methods (FSP, CP and WI types). This piece of information is important as it determines the catalyst performance in general. Three catalysts must have similar (if not identical) surface area and porosity in order to have a fair comparison on other parameters that dictate the catalytic performance.

We agree with the Reviewer and thank them for raising this relevant point. Accordingly, **we have measured the surface area and pore volume of both binary $\text{In}_2\text{O}_3\text{-ZrO}_2$ and ternary $\text{Pd-In}_2\text{O}_3/\text{ZrO}_2$ catalysts prepared by co-precipitation (CP), wet impregnation (WI), and flame spray pyrolysis (FSP), which are summarized in Supplementary Figure 9.** The results show that all systems possess similar textural properties, except for $\text{Pd-In}_2\text{O}_3\text{-ZrO}_2$, CP. The latter displays an inferior surface area compared to its counterparts, further confirming that CP is not an ideal synthesis method to produce $\text{Pd-In}_2\text{O}_3\text{-ZrO}_2$ catalysts with superior performance. Hence, this sample was considered unsuitable for a fair comparison with the other ternary systems.

5. Similar structural analysis (i.e., Supplementary Figure 10) should be done for those prepared by WI method in order to demonstrate the catalyst model described in Figure 9 (WI-derived catalyst), that is, forming the surface sheets of In_2O_3 .

A similar structural analysis was performed for the $\text{Pd-In}_2\text{O}_3/\text{ZrO}_2$ catalyst prepared by WI by microscopy techniques and has been added to Supplementary Figure 13 upon revision. The results confirm the formation of monolayer-like In_2O_3 platelets on the ZrO_2 surface.

REVIEWERS' COMMENTS

Reviewer #1 (Remarks to the Author):

Please go ahead, as a communication, I think it is worth to be published.

Reviewer #2 (Remarks to the Author):

In this revised version, the authors have largely, though not completely, addressed this reviewer's comments. The quality of the manuscript and the paper description have been improved. Considering some difficult issues that the authors are facing at this stage, therefore, the paper can be recommended for its publication in Nat. Communications. if Review 1 also accepts the authors' responses.

Manuscript NCOMMS-22-21949-A - Response to Reviewers

Notes: Comments in *blue* - Replies in black - Actions in **bold**

Indicated page, line, figure, or reference numbers refer to the revised manuscript and/or supporting information with changes highlighted

Reviewer #1

Please go ahead, as a communication, I think it is worth to be published.

We are delighted to read the Reviewer's positive feedback recognizing the impact and quality of our revised communication.

Reviewer #2

In this revised version, the authors have largely, though not completely, addressed this reviewer's comments. The quality of the manuscript and the paper description have been improved. Considering some difficult issues that the authors are facing at this stage, therefore, the paper can be recommended for its publication in Nat. Communications. if Review 1 also accepts the authors' responses.

We warmly thank the Reviewer for appreciating our efforts to tackle the Reviewers comments and understanding the current challenges. Their suggestions in the previous revision were highly valuable in allowing us to improve the current work.